# Significance of BRAF Kinase Inhibitors for Melanoma Treatment: From Bench to Bedside

**DOI:** 10.3390/cancers11091342

**Published:** 2019-09-11

**Authors:** Taku Fujimura, Yasuhiro Fujisawa, Yumi Kambayashi, Setsuya Aiba

**Affiliations:** 1Department of Dermatology, Tohoku University Graduate School of Medicine, Sendai 980-8574, Japan; yumi1001@hosp.tohoku.ac.jp (Y.K.); saiba@med.tohoku.ac.jp (S.A.); 2Department of Dermatology, University of Tsukuba, Tsukuba 305-8576, Japan; fujisan99@icloud.com

**Keywords:** BRAF-mutant metastatic melanoma, BRAF inhibitors, MEK inhibitors, immune checkpoint inhibitors, HDAC inhibitors, BRAF resistance

## Abstract

According to clinical trials, BRAF kinase inhibitors in combination with MEK kinase inhibitors are among the most promising chemotherapy regimens for the treatment of advanced BRAF-mutant melanoma, though the rate of BRAF mutation gene-bearing cutaneous melanoma is limited, especially in the Asian population. In addition, drug resistance sometimes abrogates the persistent efficacy of combined therapy with BRAF and MEK inhibitors. Therefore, recent pre-clinical study-based clinical trials have attempted to identify optimal drugs (e.g., immune checkpoint inhibitors or histone deacetylase (HDAC) inhibitors) that improve the anti-melanoma effects of BRAF and MEK inhibitors. In addition, the development of novel protocols to avoid resistance of BRAF inhibitors is another purpose of recent pre-clinical and early clinical trials. This review focuses on pre-clinical studies and early to phase III clinical trials to discuss the development of combined therapy based on BRAF inhibitors for BRAF-mutant advanced melanoma, as well as mechanisms of resistance to BRAF inhibitors.

## 1. Introduction

RAF proteins are regulators of the ERK MAP kinase signaling cascade, and interaction with RAS-GTP at the membrane promotes RAF kinase activation and leads to activation of phosphorylation of MEK1 and MEK2 [1], which play significant roles in melanoma cell proliferation, leading to high responses rates to BRAF inhibitors in melanomas compared to cytotoxic drugs [2,3,4,5,6]. Indeed, the overall response rates to vemurafenib, dabrafenib, and encorafenib monotherapies are 45%, 51%, and 60%, respectively [4,5,6]. On the other hand, resistance to BRAF kinase inhibitors is another problem in real-world practice [7,8]. Mutationally activated BRAF is expressed in several cancers, including melanoma [1], and the most common BRAF mutation leads to the substitution of a glutamic acid for valine at amino acid 600 (V600E) in the kinase domain of the protein. However, other non-V600E alterations in BRAF promote activation of other RAF forms, leading to the acquisition of resistance to BRAF-V600E inhibitors. Therefore, the anti-tumor effects of these BRAF inhibitors are enhanced by co-administration of MEK inhibitors [9,10,11], and combined therapy with a BRAF inhibitor and MEK inhibitor is recommended as one of the first-line therapies for advanced BRAF-V600-mutated melanoma.

BRAF-mutated kinase is the driver mutation found in approximately 30.4–66.0% of cutaneous melanomas [12,13]. Since large Japanese populations of melanoma subtypes are acral lentiginous melanoma (ALM) and mucosal melanoma, which have low levels of tumor mutation burden (TMB) and BRAF mutation [12,14,15], TMB might be correlated to the BRAF mutation rate. Notably, high TMB was correlated with increased neoantigens, which could be predictive markers for immune therapy, especially immune checkpoint inhibitors (ICIs) in various cancers including melanoma [16,17]. For example, in melanoma, a lower nonsynonymous mutation burden is correlated with negative results for PD-L1 expression on melanoma cells and significantly worse melanoma-specific survival in stage III melanoma [HR = 0.28 (95%CI: 0.12–0.66), *p* = 0.002] [18]. Taken together, these reports suggest the significance of assessing BRAF-mutated driver genes for not only the selection of patients who could be treated with BRAF/MEK inhibitors, but also for the prediction of the efficacy of subsequent immune therapy.

Given the above reasons, investigation of the mechanisms of the acquisition of resistance, as well as the immunological background of the tumor microenvironment, is important to establish long-term schedules for the treatment of advanced melanoma patients. Indeed, recent reports have focused on the immunomodulatory effects of BRAF/MEK inhibitors on the tumor microenvironment in melanoma-bearing hosts [19,20,21,22]. For example, Hu-Lieskovan et al. demonstrated the different immunomodulatory effects of dabrafenib and vemurafenib in the tumor microenvironment using the SM1 mouse melanoma model [19]. In addition, the functional abrogation of tumor-associated myeloid cells, such as myeloid-derived suppressor cells (MDSCs) and tumor-associated macrophages (TAMs), have been shown to improve the anti-melanoma effects of BRAF/MEK inhibitor-resistant melanoma [22,23]. Since TAMs develop sequentially from monocytes into functional macrophages and can obtain various immunosuppressive functions by the stimulation of cancer stromal factors in each differentiation stage [24], the immunomodulation of TAMs might enhance the anti-tumor effects of various drugs including BRAF/MEK inhibitors [22]. In addition to TAMs, the assessment of CD8+ effector T cells is important to augment the anti-tumor response in BRAF inhibitor-resistant melanomas [20,21]. Taken together, the phenotypic analysis of tumor-infiltrating leukocytes (TILs) might be important not only to evaluate the efficacy of immune therapy [25,26,27], but also to understand the mechanisms to overcome the acquisition of resistance to BRAF inhibitors [20,21,22,23].

## 2. Methods

### Search Strategy for the Literature Search

Areas covered: The literature review was performed on PubMed (search period: From 1950 to 2019/8/17) to identify these drugs using search terms such as ‘melanoma’, ‘BRAF inhibitors’, ‘MEK inhibitors’, ‘phase 1 clinical trial’, ‘phase 2 clinical trial’, ‘phase 3 clinical trial’, ‘immunological background’, ‘tumor-infiltrating leukocytes’, ‘immune checkpoint inhibitors’, ‘adverse events’, and ‘drug resistant’. The results of the literature search with relevant keywords are summarized in Table 1.

## 3. Pre-Clinical Investigations of BRAF Kinase Inhibitors

As described above, resistance to BRAF kinase inhibitors is important for the long-term treatment of advanced melanoma in the real world [7,8]. Therefore, several pre-clinical studies to investigate the mechanisms of drug resistance have recently been reported [28,29] (Table 2).

For example, Thakur et al. reported that vemurafenib-resistant melanoma cells in a xenograft mouse model showed continuous vemurafenib-dependent proliferation that was recovered by stopping drug administration, and altered dosing might prevent the emergence of lethal drug resistance [30]. This report might be useful to improve vemurafenib therapy by sustaining the durability of the vemurafenib response. Today, a phase II clinical trial of vemurafenib plus cobimetinib continuous vs intermittent in BRAF V600-mutant melanoma is ongoing (NCT02583516). In another report, Corre et al. reported that aryl hydrocarbon receptor (AhR) transcription factor constitutively activates human melanoma cells, which inhibit the differentiation of melanoma cells and express BRAF inhibitor-resistance genes, such as MITF-targets and pigmentation gene, that overlap with a classical proliferative signature [28]. Notably, they also demonstrated that both vemurafenib and dabrafenib bind directly to AhR β-pockets, which are different binding sites of canonical AhR ligands, such as TCDD and FICZ [28]. Furthermore, this study also suggested that targeting AhR signaling could prevent the induction of the BRAF inhibitor resistance gene in melanoma cells, thus augmenting the efficacy of BRAF inhibitors [28]. Since numerous AhR ligands have been reported and further studies are needed to apply their findings directly to the bedside, their findings might alter BRAF inhibitor resistance in the future. In another report, Wang et al. reported that resistance to BRAF plus MEK inhibitors combination therapy is associated with reactive oxygen species (ROS) activities in human melanoma cell lines [31]. Interestingly, they suggested that the administration of a histone deacetylase (HDAC) inhibitor induces selective apoptotic death of drug-resistant tumor cells. They concluded that the HDAC inhibitor, vorinostat, might prolong the anti-tumor effects of BRAF/MEK inhibitor combination therapy [31]. Concerning the HDAC inhibitor as a supportive drug for resistance to BRAF inhibitors, a combination of a broad-spectrum HDAC inhibitor (e.g., quisinostat) and a pan-CDK inhibitor (e.g., flavopiridol) has been reported to synergistically suppress the proliferation of BRAF inhibitor-resistant human melanoma cells in a xenograft mouse model [32]. In another report, panobinostat recovered the anti-melanoma effects of encorafenib in BRAF inhibitor-resistant melanoma cell lines by the induction of caspase-dependent apoptotic cell death in human melanoma cell lines [33]. Based on these findings, phase I/II clinical studies using the HDAC inhibitor vorinostat in resistant BRAF V600-mutated advanced melanoma have been performed in the Netherlands. Furthermore, more recently, Gupta et al. reported that the lack of block of proliferation 1 (BOP1) decreases the MAP phosphatases, dual specificity phosphatase (DUSP)4, and DUXP6, leading to increased MAP signaling and BRAF resistance in three human melanoma cell lines (SKMEL-239, SKMEL-28, A375) [29]. Taken together, these reports suggest the mechanisms of BRAF inhibitor resistance and possible ways to improve the treatment of BRAF inhibitor-resistant advanced melanoma.

Although the response rate to BRAF inhibitor plus MEK inhibitor combination therapy is as high as 67% [4,11,36], median progression-free survival (PFS) is approximately 12 months [11,36], suggesting the need for additional promising methods that could prolong anti-melanoma effects. Therefore, several investigators have sought the immunomodulatory drugs that could modulate the cancer stroma for the optimal immune therapy [19,22,37,38]. For example, Mok et al. reported that the colony stimulating factor (CSF)-1 receptor inhibitor PLX3397 enhances the anti-melanoma effects of vemurafenib by decreasing tumor-infiltrating myeloid cells [22]. Notably, compared to dabrafenib, the immunomodulatory effects of vemurafenib on myeloid cells are limited [19,38]. Since CSF-1 is needed to polarize the TAMs into the immunosuppressive M2 phenotype at the first stage of their generation [37], the blockade of CSF-1 could decrease the TAMs at the tumor sites, leading to induction of immunoreactive effector T cells at the tumor site [22]. In addition, Steinberg et al. reported that since MDSCs could be associated with resistance to BRAF inhibitors [23], the depletion of MDSCs enhances the efficacy of anti-PD1 antibody plus anti-CTLA4 antibody combination therapy [23]. This report also suggested the significance of targeting myeloid cells in the tumor microenvironment.

Both myeloid cells and CD8+ effector cells play important roles in the induction of resistance to BRAF inhibitors. Therefore, several reports focused on the effects of BRAF inhibitors on the induction of CD8+ T cells [21,34,35]. For example, Cooper et al. reported that the administration of BRAF inhibitors increases the CD8+ T cells among tumor-infiltrating lymphocytes (TILs) and decreases immunosuppressive cytokines, leading to suppression of growth of BRAF (V600E)/PTEN−/− melanoma in vivo [34]. In another report, the additional administration of vemurafenib to an adoptive transfer of autologous TILs significantly enhanced the anti-tumor effects of patients’ TILs in BRAF-resistant tumor-bearing xenograft mice [21]. Moreover, Homet Moreno et al. reported that the anti-tumor effects of dabrafenib plus trametinib combination therapy could be enhanced by immunostimulatory antibodies (Abs) (e.g., 4-1BB Abs, OX40 Abs) [35]. Indeed, dabrafenib plus trametinib combination therapy together with anti-PD1 Abs increased the population of CD8+ T cells among TILs, enhancing the anti-tumor activity that was further improved by anti-4-1BB Abs [35]. These reports suggest that the efficacy of BRAF inhibitors could be improved by combining them with immunotherapy, such as ICIs, to exert long-acting anti-tumor immune responses.

## 4. Clinical Trials of BRAF Kinase Inhibitors in the Treatment of Melanoma

Since there are several promising pre-clinical studies of the treatment of advanced melanoma, clinical studies are needed to translate these pre-clinical data into treatments for patients with advanced melanoma. In this section, we describe already published phase I, phase II, and phase III clinical studies, as well as novel clinical studies using BRAF inhibitor-based combination therapies with other types of drugs in the treatment of advanced melanoma.

### 4.1. Phase I Studies

To date, there are three different BRAF inhibitors available for advanced melanoma treatment: Vemurafenib, dabrafenib, and encorafenib. All BRAF inhibitors are currently used as combined therapy with MEK inhibitors that can not only improve the anti-tumor effects (e.g., response rate or duration of response), but can also reduce the occurrence of squamous cell carcinoma, which is a commonly seen severe adverse event when a BRAF inhibitor is used as monotherapy [4,11,36].

Although the response rate to the current combined therapies using BRAF and MEK inhibitors is as high as 67% [4,11,36], most patients relapse even with this combined therapy. Such tumors acquire resistance by re-activation of the MAPK pathway through several distinct mechanisms, such as mutation of an N-RAS mutation, COT1 accumulation [39], or acquired MEK2 mutation [40]. Targeting heat shock protein 90 (HSP90), a molecular chaperone that controls correct folding and stability of the BRAF mutant protein, has shown activity in pre-clinical models of melanoma including vemurafenib resistance [41,42]. Indeed, a clinical trial using XL888, an HSP90 inhibitor, combined with vemurafenib in participants with BRAF V600-mutated melanoma achieved 3 complete and 12 partial responses in 20 evaluable patients (response rate, 75%), with median progression-free and overall survivals of 9.2 months and 34.6 months, respectively [43]. In this study, the dose of vemurafenib was determined to be 960 mg (twice daily), and that of XL888 was 90 mg (once a day), since three dose-limiting toxicities (DLTs) were observed in the next escalation cohort (XL888, 135 mg). However, since BRAF inhibitors are currently accepted as combined therapy with a MEK inhibitor to gain a higher response and reduced toxicity, a phase I clinical trial to determine the maximum tolerated dose (MTD) and recommended phase II dose of XL888 when administered orally with vemurafenib plus cobimetinib and to evaluate the safety and tolerability of this combination is currently ongoing (NCT02721459).

One of the mechanisms of resistance to BRAF inhibition occurs through the upregulation of pro-survival signaling pathways, such as the phosphoinositide-3-kinase (PI3K) pathway [44]. Based on this mechanism, a phase I clinical trial of PX866, a potent PI3K inhibitor, combined with vemurafenib has been conducted [45]. Since the MTD of single-agent PX866 was determined to be 8 mg per day in the single-agent dose-escalation phase I study by Hong et al. [46], four cohorts were tested by combining a low dose and a high dose of PX866 with vemurafenib (PX866, 6 mg and 8 mg per day and vemurafenib, 720 mg or 960 mg twice daily). It was found that 8 mg of PX866 was well-tolerated with 720 mg of vemurafenib, and 7 of 24 patients (29.2%) achieved a clinical response. Notably, 10 of 24 patients received prior BRAF or MEK inhibition therapy before entering this trial, suggesting that this combination might be active for BRAF-resistant patients.

Other possible resistance mechanisms to BRAF inhibition could occur through high BCL2 expression by tumor cells, leading to resistance to BRAF inhibition-induced apoptosis [47]. Combined with the use of a BRAF inhibitor, navitoclax, a potent BCL2-inhibitor, acts synergistically to reduce the viability of BRAF mutant cell lines and xenograft models [48]. Based on this result, a phase I study using navitoclax combined with dabrafenib and trametinib (MEK inhibitors) in patients with BRAF mutant carcinoma (meaning not limited to melanoma) was conducted (NCT01989585). The result of this study was presented at the 2018 American Association for Cancer Research Conference (LB-B30, available online, http://mct.aacrjournals.org/content/17/1_Supplement/LB-B30). Out of 11 patients with advanced melanoma, 5 of them had prior BRAF inhibitor therapy, and 6 of them were BRAF inhibitor-naïve. Interestingly, all patients without prior BRAF inhibitor therapy responded (one complete and five partial responses), whereas zero of six patients who had prior BRAF inhibitor therapy responded. The recommended navitoclax dose for a phase II study was determined to be 225 mg with the standard dose of dabrafenib (150 mg, twice daily) and trametinib (2 mg per day). The randomized phase II portion of this study to compare dabrafenib plus trametinib with or without navitoclax in BRAF inhibitor therapy-naïve cancer patients is currently ongoing.

Chemotherapy combined with BRAF inhibition was also tested in phase I studies. Bhatty et al. conducted a phase I trial to determine the safe dose of vemurafenib, as well as carboplatin/paclitaxel, in patients with BRAF-mutated tumors [49]. In this trial, the MTD was not reached, and vemurafenib at 720 mg twice daily, carboplatin at AUC5, and paclitaxel at 135 mg/m^2^ were the last safe dose levels. Although 13 of 19 evaluable patients were previously treated with BRAF and/or MEK inhibitors, 26% patients achieved clinical response (one CR and four PR patients).

Another cytotoxic agent, decitabine, a DNA methyltransferase inhibitor, has been reported to induce apoptosis and have immunomodulatory effects. Decitabine has previously been tested in combination with several anti-melanoma drugs with promising results [50,51]. Based on these results, Zakharia et al. [52] conducted a phase Ib trial using decitabine combined with vemurafenib and reported 3 complete and 3 partial responses in 14 patients (response rate: 42.8%). No dose-limiting toxicity due to the treatment was detected.

Since most clinical studies included patients 18 years old or older, we do not have evidence to show that the dose of current BRAF inhibitors is sufficient for treating patients younger than 18 years. Chisholm et al. [53] conducted a phase I dose-escalation study using a 3 + 3 design (starting with 720 mg, twice daily) to determine the MTD by enrolling patients 12–17 years old with melanoma with BRAF mutation. In this study, all patients experienced at least one adverse event (AE) and three patients (60%) experienced serious AEs, whereas zero of six evaluable patients had no objective response, with a median PFS of 4.4 months. Since the number of patients enrolled in this study was very small (*n* = 6), it is difficult to determine the effective and recommended dose of vemurafenib.

### 4.2. Phase II Studies

A phase II clinical trial of vemurafenib plus cobimetinib continuous vs, intermittent in BRAF V600-mutant melanoma is currently ongoing (NCT02583516) to evaluate the alternative dosing regimens of MAPK pathway inhibition according to their pre-clinical studies [30]. According to this clinical study, the protocol for the administration of vemurafenib plus cobimetinib combination therapy might be improved to overcome drug resistance.

Although MEK inhibitors are the current best partner for BRAF inhibitor therapy, many other drugs in combination with BRAF and MEK inhibitors are now in clinical trials. Of those, the use of immune checkpoint inhibitors (ICIs) is thought to be one of the best candidates, since the dynamics and the durability of response differ greatly, with a high response rate and short response time, but a high rate of drug resistance with a BRAF inhibitor, compared to a lower response rate and longer response time, but a durable response with ipilimumab [54]. Moreover, the use of BRAF inhibitors could lead to increased expression of melanocyte differentiation antigens and CD8+ T cell tumor infiltration and decreased immunosuppressive cytokines [55,56], all of which can enhance the efficacy of immunotherapy. In this context, a combination of BRAF inhibitor (vemurafenib) and ICI (ipilimumab) has been tested, but the study was terminated due to a high rate of severe hepatotoxicity [57]. Thus, Amin et al. conducted a phase II study of sequential administration of vemurafenib followed by ipilimumab to avoid toxicity [58]. Indeed, the toxicity was manageable, but the outcome was controversial. In addition, high-dose interleukin-2 therapy also served as one of the immunotherapies for melanoma treatment, and several trials were performed in combination with BRAF inhibitors. Although the combination of high-dose interleukin-2 and vemurafenib did not change the known toxicity profile for either agent, the response rate was lower than expected [59,60], and the role of combination high-dose interleukin-2 and vemurafenib remains uncertain. Overall, these phase II studies suggest that BRAF inhibitors should be combined with MEK inhibitors, even as sequential or combined therapy with immunotherapy.

Several phase II studies evaluating combined BRAF and MEK inhibitors with immunotherapy such as anti-PD-1 antibody are currently underway. Among them, for example, a phase II study to evaluate the efficacy and toxicity of sequential administration of nivolumab and ipilimumab after vemurafenib and cobimetinib is ongoing (NCT02968303). Moreover, the result of a randomized phase II study evaluating pembrolizumab and anti-programmed death-1 antibody (aPD-1), combined with dabrafenib and trametinib for BRAF-mutant advanced melanoma, was reported recently [61]. Patients in the triple-therapy arm had numerically longer PFS than the placebo plus dabrafenib and trametinib arm, but the difference was not significant (median 16.0 and 10.3 months, respectively). Overall survival (OS) and best overall response were similar between the treatment arms. The rate of serious AEs was higher in the triplet treatment arm, and one treatment-related death occurred because of pneumonitis. Since the evaluated number of patients was low in this phase II study [61], the role of an anti-PD-1 inhibitor in combination with BRAF and MEK inhibitors should be validated in a further phase III study.

Brain metastasis is known to correlate with poor survival, with a median survival of 4–5 months [62]. BRAF inhibition was tested in this poor outcome population as BRAF inhibitor monotherapy [63] and BRAF and MEK inhibitor combination therapy [64]. With BRAF inhibitor monotherapy, vemurafenib achieved a 26% response rate, including one complete response, with a median OS of 8.9 months [63]. On the other hand, BRAF and MEK inhibitor combination therapy achieved higher efficacy, with a response rate of 44% to 65% and a median survival of 10.8 months [64], which was better than vemurafenib monotherapy. Overall, these reports suggested that BRAF and MEK inhibitor combination therapy is one of the optimal therapies for metastatic melanoma in the brain. Since the evaluated number of patients was low, these findings should be validated with a large cohort in a further phase III study.

Another interesting trial, reported by Amaria et al., evaluated the use of BRAF and MEK inhibitors as neoadjuvant and adjuvant therapies for the treatment of high-risk, surgically resectable stage 3 melanoma [65]. This phase II randomized trial showed that patients with pre-use of BRAF and MEK inhibitors had significantly improved relapse-free survival compared with standard care.

### 4.3. Phase III Studies

Based on several prospective phase III studies, dabrafenib plus trametinib is widely used today for the treatment of BRAF^V600^-mutated melanoma. A summary of the phase III studies for BRAF inhibitors plus MEK inhibitors is shown in Table 3 and Figure 1. The analysis of COMBI-d (NCT01584648) demonstrated improved PFS and OS with dabrafenib plus trametinib combination therapy compared with dabrafenib monotherapy [5,10]. The landmark analysis of this study is described below: Three-year PFS was 22% for dabrafenib plus trametinib combination therapy and 12% for the dabrafenib monotherapy arm [HR: 0.71: (95% CI, 0.57–0.88)], and three-year OS was 44% and 32%, respectively [HR: 0.75, (95% CI, 0.58–0.96)] [10]. Since the comparison of other BRAF inhibitors is important to evaluate the efficacy of dabrafenib plus trametinib combination therapy, another phase III study, COMBI-v (NCT01597908), demonstrated [2,3,36] that one-year OS was 72% for dabrafenib plus trametinib combination therapy and 65% for the vemurafenib monotherapy arm [HR: 0.69, (95% CI, 0.53–0.89)] [36]. The median PFS was 11.4 months in the dabrafenib plus trametinib combination therapy group and 7.3 months in the vemurafenib group [HR: 0.56, (95% CI, 0.46–0.69, *p* < 0.001)]. Notably, Schadendorf et al. reported a three-year pooled analysis of the combined COMBI-v and COMBI-d trials [66], suggesting that the predictive markers for PFS and OS are the baseline LDH level and number of organ sites involved. Indeed, the group with normal LDH, baseline sum of lesion diameters (SLD) <66 mm, and <3 organ sites involved had the most favorable outcomes when treated with dabrafenib plus trametinib combination therapy (three-year PFS 42%, three-year OS 55%) [66]. This study supports selecting patients who should be treated by dabrafenib plus trametinib combination therapy. These findings suggest that the response rate of dabrafenib plus trametinib combination therapy is sufficient, but the OS and PFS are not sufficient, indicating that other drugs that enhance dabrafenib trametinib therapy are needed.

Dabrafenib plus trametinib combination therapy is not only useful for unresectable advanced melanoma but is also useful in the adjuvant setting [67]. The analysis of COMBI-AD (NCT01682083) demonstrated improved relapse-free survival in the combined therapy group (HR for relapse or death, 0.47; 95%CI, 0.39–0.58; *p* < 0.001). The three-year OS rate was 86% in the combined therapy group and 77% in the placebo group (HR, 0.57; 95% CI, 0.42–0.79; *p* = 0.0006).

The analysis of COLUMBUS (NCT01909453) demonstrated improved PFS and OS with encorafenib plus binimetinib combination therapy and encorafenib monotherapy compared with vemurafenib monotherapy [11]. Notably, unlike dabrafenib and trametinib combination therapy, although OS was significantly improved in both encorafenib plus binimetinib combination therapy (encorafenib plus binimetinib vs. vemurafenib: [HR 0.61: 95%CI 0.47–0.79, *p* < 0.0001]) and encorafenib monotherapy (encorafenib vs. vemurafenib: [HR 0.76: 95%CI 0.58–0.98, *p* < 0.033]) compared with vemurafenib, there was no significant difference between encorafenib plus binimetinib combination therapy and encorafenib monotherapy (encorafenib plus binimetinib vs. encorafenib: [HR 0.81: 95%CI 0.61–1.06, *p* = 0.12]). In addition, PFS was significantly improved in both encorafenib plus binimetinib combination therapy (encorafenib plus binimetinib vs. vemurafenib: [HR 0.51: 95%CI 0.39–0.67, *p* < 0.0001]) and encorafenib monotherapy (encorafenib vs. vemurafenib: [HR 0.68: 95%CI 0.52–0.88, *p* = 0.0038]) compared with vemurafenib. Unlike OS, PFS was significantly improved in encorafenib plus binimetinib combination therapy compared with encorafenib monotherapy (encorafenib plus binimetinib vs. encorafenib: [HR 0.77: 95%CI 0.59–1.00, *p* = 0.050]). The median OS for encorafenib plus binimetinib vs encorafenib vs. vemurafenib was 33.6 (95% CI: 24.4–39.2), 23.5 (95% CI: 19.6–33.6), and 16.9 (95% CI: 14.0–24.5) months, respectively, and the median PFS for encorafenib plus binimetinib vs encorafenib vs. vemurafenib was 14.9 (95% CI: 11.0–20.2), 9.6 (95% CI: 7.4–14.8), and 7.3 (95% CI: 5.6–7.9) months, respectively.

Vemurafenib plus cobimetinib combination therapy also improves PFS compared with vemurafenib monotherapy [9]. The analysis of coBRIM (NCT01689519) demonstrated improved PFS and OS with vemurafenib plus cobimetinib combination therapy compared with vemurafenib monotherapy [9]. PFS was significantly improved with vemurafenib plus cobimetinib combination therapy (HR 0.58: 95%CI 0.46–0.72, *p* < 0.0001). OS was also significantly improved with vemurafenib plus cobimetinib combination therapy, though the statistical significance was low (HR 0.70: 95%CI 0.55–0.90, *p* = 0.005).

### 4.4. Adverse Events

As described above, BRAF/MEK inhibitors could have an effect on the immunological background of the tumor microenvironment, and these immune reactions play a role in developing characteristic AEs. Compared with cytotoxic drugs, such as dacarbazine (DTIC), a high rate of various serious adverse events (SAEs) has been reported in patients treated with BRAF inhibitor monotherapy and BRAF plus MEK inhibitor combination therapy [5,6,7,9,10,11,25]. One of the possible explanations for the high rates of cutaneous toxicity caused by BRAF inhibitor monotherapy might be paradoxical activation of the MAPK kinase pathway in keratinocytes, leading to a higher incidence of cutaneous keratotic tumors such as keratoacanthoma and squamous cell carcinoma (SCC) [5,6,9,10,11] that could logically be suppressed by adding a MEK inhibitor [5,6,7,9,10,11]. Interestingly, there is a single case report that described vemurafenib-induced multiple keratoacanthoma and palmoplantar keratoderma that was rapidly diminished by switching to dabrafenib and trametinib combination therapy [27]. This case report suggested that BRAF induced keratoacanthoma directly.

In addition, the subtypes of SAEs differed between BRAF inhibitor monotherapy and BRAF plus MEK inhibitor combination therapy [9,10,11], suggesting that BRAF inhibitors and MEK inhibitors lead to the development of independent AEs with different pathogeneses. For example, the incidence of severe pyrexia, chills, diarrhea, and vomiting that might be caused by MEK inhibitors is higher with combined therapy, whereas tumor development, such as keratoacanthomas, SCC, and palmoplantar hyperkeratosis, is higher with monotherapy [4,10,67]. Moreover, a high frequency of serous retinitis is observed in patients with both encorafenib monotherapy and encorafenib and binimetinib combination therapy [11], whereas severe pyrexia is observed in patients with dabrafenib and trametinib combination therapy [9,10]. These reports suggest that the subtypes of AEs seen are also different among the BRAF inhibitors vemurafenib, dabrafenib, and encorafenib [9,10,11].

Sequential administration of BRAF inhibitors with anti-PD1 antibodies might enhance the incidence of unexpected SAEs in the clinical setting [68]. For example, there are several case reports that described SAEs, such as Vogt-Koyanagi Harada disease-like uveitis and severe rhabdomyolysis, in advanced melanoma patients with nivolumab monotherapy followed by BRAF/MEK inhibitor combination therapy [69,70,71]. Moreover, nivolumab followed by dabrafenib plus trametinib therapy can cause severe drug eruptions, such as erythema exudative multiforme [58]. These reports suggest that the actual immune-related AEs might be different from those described in previously published clinical studies.

## 5. Future Perspective

Although BRAF plus MEK inhibitor combination therapy rapidly suppresses melanoma growth, the recurrence rate following these combined therapies is a concern [9,10,11], suggesting the necessity of additional therapy that could induce long-acting anti-melanoma effects. Today, one of the promising methods for the induction of long-acting anti-tumor effect is the use of ICIs. Indeed, ICIs prolonged OS and PFS in patients with metastatic melanoma [36,72]. Notably, the recent pre-clinical studies suggested possible novel therapies for the treatment of advanced melanoma using combined therapy of BRAF/MEK inhibitors and ICIs [19,35,38]. Indeed, several phase I/II clinical studies for the treatment of patients with advanced melanoma have been set up based on these pre-clinical studies. Among them, a major phase I study (NCT02130466, NCT02818023) and a phase II study (NCT02968303, NCT03514901) for BRAF inhibitors in combination with anti-PD1 antibody are ongoing. In addition, a phase II study to evaluate the efficacy and toxicity of sequential administration of nivolumab and ipilimumab after vemurafenib and cobimetinib is also ongoing (NCT02968303). Moreover, the result of a randomized phase II study evaluating pembrolizumab combined with dabrafenib and trametinib for BRAF-mutant advanced melanoma was reported recently [61]. Since the evaluated number of patients was low in this phase II study [61], the role of an anti-PD-1 inhibitor in combination with BRAF and MEK inhibitors should be validated in a further phase III study. Unlike anti-PD1 antibodies, another ICI, ipilimumab, is unsuitable for combined therapy with BRAF inhibitors because of high rates of severe hepatotoxicity (NCT01400451) [57,58]. Overall, BRAF/MEK inhibitor-based combined therapies with ICIs might be next-generation therapies for BRAF-mutant advanced melanoma in the near future.

As described above, BRAF inhibitors should be combined with MEK inhibitors for developing novel combination therapies. ICIs in combination with BRAF and MEK inhibitors, as well as HDAC inhibitors, are possible in combination with BRAF/MEK inhibitors and might prolong the anti-tumor effects of BRAF/MEK inhibitor combination therapy [31,32]. Indeed, since phase I/II clinical studies using the HDAC inhibitor vorinostat in resistant BRAF V600-mutated advanced melanoma have already been performed, the combined therapy of BRAF/MEK inhibitors with HDAC inhibitors in a large cohort should be explored in the near future.

The development of BRAF and MEK inhibitors with other combined drugs and the investigation of the appropriate doses to avoid drug resistance should serve as interests for oncologists. A pre-clinical study using a xenograft mouse model showed that cessation of vemurafenib could prevent the emergence of lethal drug resistance [30]. Therefore, vemurafenib plus cobimetinib combination therapy, dabrafenib plus trametinib combination therapy, and encorafenib plus binimetinib combination therapy might improve these combined therapies by sustaining the durability of the BRAF and MEK inhibitor response. Overall, the development of BRAF/MEK inhibitor-based combined therapies could prolong patients’ OS.

## Figures and Tables

**Figure 1 cancers-11-01342-f001:**
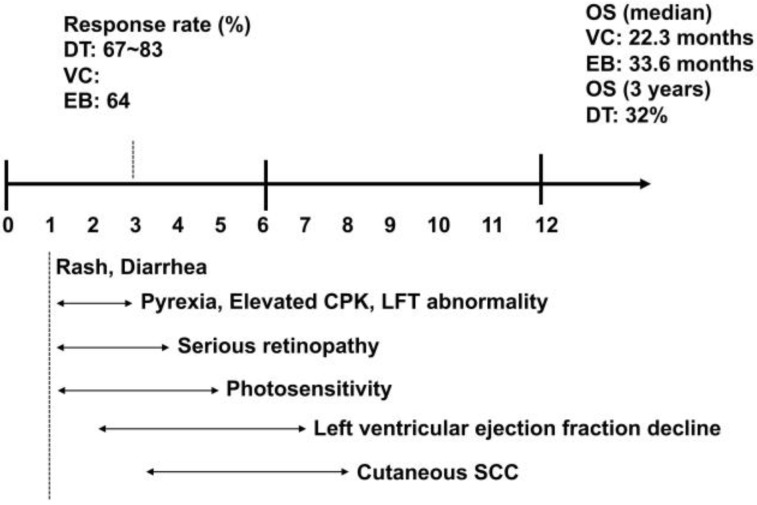
Efficacy and adverse events of combined therapy of BRAF inhibitors plus MEK inhibitors. DT: Dabrafenib plus trametinib therapy; VC: Vemurafenib plus cobimetinib therapy; EB: Encolafenib plus binimetinib therapy; OS: Overall survival; LFT: Liver function test; SCC: Squamous cell carcinoma.

**Table 1 cancers-11-01342-t001:** Search Strategy for the Literature Search.

#No.	Retrieval Style	No. of References
#01	“Melanoma/ BRAF or MEK” [TIAB]	19,869
#02	“Melanoma/ BRAF” [TIAB] or “BRAF inhibitors” [TIAB]	5929
#03	“Melanoma/ BRAF inhibitors” [TIAB] and/ or “MEK inhibitors” [TIAB])	10,517
#04	“Melanoma/ BRAF inhibitors” [TIAB] and/ or “MEK inhibitors and mouse” [TIAB]	4957
#05	“Melanoma/ BRAF inhibitors” [TIAB] and “mouse” [TIAB]	471
#06	“Melanoma/ BRAF inhibitors” [TIAB] and “clinical trials” [TIAB]	344
#07	“Melanoma/ BRAF inhibitors” [TIAB] or “MEK inhibitors” [TIAB] and “adverse event” [TIAB]	2613
#08	“Melanoma/ BRAF inhibitors” [TIAB] and “adverse events” [TIAB]	54
#09	“Melanoma/ BRAF inhibitors” [TIAB] and “clinical trials” [TIAB] and “adverse events” [TIAB]	17
#10	“Melanoma” [TIAB] and “immunological background” [TIAB]	214
#11	“Melanoma” [TIAB] and “tumor infiltrating leukocytes” [TIAB]	1793
#12	“Melanoma” [TIAB] and “immune checkpoint inhibitors” [TIAB]	1439
#13	“Melanoma/ BRAF inhibitors” [TIAB] or “MEK inhibitors” [TIAB] and “drug resistance” [TIAB]	373

**Table 2 cancers-11-01342-t002:** Pre-clinical trials for BRAF-mutant melanoma.

Category	Target Molecules	Output	Target Cells	Ref.
Drug resistance	aryl hydrocarbon receptor (AhR)	differentiation	melanoma cells	[28]
	reactive oxygen species (ROS)	induction of apoptosis	drug-resistant tumor cells	[31]
	HDAC, pan-CDK	reduce cell viability	drug-resistant tumor cells	[32]
	block of proliferation 1 (BOP1)	decrease MAP signaling	drug-resistant in melanoma cells	[29]
Immune regulation	CSF-1 receptor	reduction of myeloid cells	tumor-infiltrating myeloid cells	[22]
	cell growth	suppress tumor growth	CD8+ T cells	[34]
	4-1BB, OX40	suppress tumor growth	CD8+ T cells	[35]

**Table 3 cancers-11-01342-t003:** Summary of the efficacies of BRAF inhibitor monotherapy and combined therapy.

Protocol	Efficacy (%)	Median OS (Months)	Median PFS (Months)	Ref.
vemurafenib monotherapy	48~52	13.6~17.4	6.9~7.3	[2,3,9,36]
dabrafenib monotherapy	50~51	13.6	5.1	[2,3,5,10,36]
encorafenib monotherapy	41	23.5	7.3	[11]
dabrafenib + trametinib combined therapy	67~83		11.4	[2,3,5,10,36]
vemurafenib + cobimetinib combined therapy		22.3	12.4	[9]
encorafenib + binimetinib combined therapy	64	33.6	19	[11]

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
