# Peer review of "Significance of BRAF Kinase Inhibitors for Melanoma Treatment: From Bench to Bedside"

_cancers, 2019, doi:10.3390/cancers11091342_

Round 1

Reviewer 1 Report

The authors have made appropriate efforts to address reviewer comments which are reflected in the revised manuscript.

Reviewer 2 Report

The authors have clearly improved the manuscript compared to the original version. There are already several other reviews in this field but in general the topic is interesting and important.

Reviewer 3 Report

Manuscript has been significantly improved with indicated changes.

This manuscript is a resubmission of an earlier submission. The following is a list of the peer review reports and author responses from that submission.

Round 1

Reviewer 1 Report

The authors have submitted an important review article that provides an update on recent clinical trials evaluating BRAF inhibitor combination therapies for melanoma. The review also includes perspectives on therapeutic pitfalls and insight on the future direction of treatment approaches.

The following comments would help to improve the manuscript.

The introduction would benefit from an expanded description of the original development of BRAF inhibitors and a somewhat more detailed explanation of their molecular mechanisms. Moreover, inclusion of a timeline figure outlining the use of BRAF inhibitors with combination therapies highlighting important results and adverse outcomes would provide a helpful supplement to the information and summarize key findings for the reader.

The detail of the methods section needs to be expanded beyond the listed search terms to be more in line with standard guidelines for reporting web-based search analysis. This includes aspects related to date searched, date accessed, number of search results, search limits, search strategy, url, etc. The authors also indicate that other resources were used in addition to PubMed which should be listed.

The discussion of the results for some clinical trials is short and should be expanded to provide more insight to the studies. Additionally, the abstract should be expanded.

Minor grammatical and spelling errors should be corrected including lines 54, 95 and 360.

Author Response

Reviewer 1

The authors have submitted an important review article that provides an update on recent clinical trials evaluating BRAF inhibitor combination therapies for melanoma. The review also includes perspectives on therapeutic pitfalls and insight on the future direction of treatment approaches.

The following comments would help to improve the manuscript.

The introduction would benefit from an expanded description of the original development of BRAF inhibitors and a somewhat more detailed explanation of their molecular mechanisms. Moreover, inclusion of a timeline figure outlining the use of BRAF inhibitors with combination therapies highlighting important results and adverse outcomes would provide a helpful supplement to the information and summarize key findings for the reader.

Thank you for your comment. We have expanded introduction as you pointed out,.

The detail of the methods section needs to be expanded beyond the listed search terms to be more in line with standard guidelines for reporting web-based search analysis. This includes aspects related to date searched, date accessed, number of search results, search limits, search strategy, url, etc. The authors also indicate that other resources were used in addition to PubMed which should be listed.

Thank you for your comments. We added additional table as you pointed out.

The discussion of the results for some clinical trials is short and should be expanded to provide more insight to the studies. Additionally, the abstract should be expanded.

Thank you for your comment. We added more information as suggested.

Minor grammatical and spelling errors should be corrected including lines 54, 95 and 360.

Thank you for your comments. We corrected the minor grammatical and spelling errors.

Reviewer 2 Report

BRAF kinase inhibitors have become an integral part of melanoma management, thus the review addresses relevant issues. On the other hand, several recent reviews have looked at resistance mechanisms and combination therapies of BRAF inhibitors and the current article adds only little information that has not already been reviewed and discussed elsewhere. An overlapping set of authors has recently published a review on the topic that shows a partial overlap with the current article.

The review is far from comprehensive and it is not always clear why specific resistance mechanisms or combination strategies are included or ignored.

A more systematic, mechanisms-oriented approach would have made the article an easier read.

Large parts of the review merely list results of clinical studies without a lot of discussion or expert opinion provided by the authors.

English is understandable but considerable language editing is still required

Overall, the manuscript is of limited relevance and quality

Author Response

As you and other reviewers pointed out, we have re-written the manuscript to address the issues raised.

Reviewer 3 Report

The authors should be commended on trying to summarise and present a very broad amount of literature on BRAF+MEK inhibitors. While there are many interesting aspects, I do feel the manuscript needs to be clearer in its aims and employed methodology. With a clearer aim for the review some information / sections may become redundant.

Introduction

·         I found it difficult to follow paragraph 1. I am unsure it adequately reflects recommended guidelines for the use of BRAF+MEK inhibitors in melanoma treatment. It may be prudent to outline how BRAF+MEK inhibitors are used in advanced melanoma, and its uses in early stages of disease. Are there differences between guidelines in Japan, and ASCO / ESMO?

·         The introduction does not outline a clear aim for the review, and the search strategy (which I found vague [e.g. use of the term ‘mainly’]) appears to reflect a very unstructured nature to process and purpose.

Abstract

·         Again, is the abstract in alignment with ASCO / ESMO guidelines for the use of BRAF+MEK inhibitors in melanoma. The aim of the review needs to more clearly stated.

Pre-clinical studies

·         There is a lot of important information presented, and the authors should be commended on trying to accumulate this evidence. But is it section truly complete? (e.g. https://www.ncbi.nlm.nih.gov/pubmed/23302800)

·         This section requires clarity that the data is pre-clinical – often without independent validation. Which studies have been repeated or the marker investigated in human data? Which have stimulated human clinical trials? Some sentences are written as if the markers are known drivers of resistance within a clinical setting, without validation, I am not sure this is the case.

Phase 1,2 ,3 studies

·         Again, there is a lot of important information presented. But is it truly complete? https://ascopubs.org/doi/abs/10.1200/JCO.2017.35.15_suppl.TPS9599

·         A structured search of clinicaltrials.gov may enable clearer and more complete outline of completed clinical trials, as well as trials that are in progress.

·         Is summarising the Phase 1, 2, 3 studies the major purpose (both completed and underway) of this review?

Adverse events

·         I question if this section belongs in a review centred on outlining clinical studies / trials conducted on BRAF+MEK inhibitors – an entire review could be written on this subject, and there are examples.

Future directions

·         The summary of human clinical trials that are currently being conducted needs to be expanded – and hopefully the pre-clinical data will provide some interesting context as to why they are being conducted.

Author Response

Reviewer 3

The authors should be commended on trying to summarise and present a very broad amount of literature on BRAF+MEK inhibitors. While there are many interesting aspects, I do feel the manuscript needs to be clearer in its aims and employed methodology. With a clearer aim for the review some information / sections may become redundant.

Thank you for your comments. As you and reviewer 1 pointed out, we added additional table to clarify the methodology in our present manuscript.

Introduction

I found it difficult to follow paragraph 1. I am unsure it adequately reflects recommended guidelines for the use of BRAF+MEK inhibitors in melanoma treatment. It may be prudent to outline how BRAF+MEK inhibitors are used in advanced melanoma, and its uses in early stages of disease. Are there differences between guidelines in Japan, and ASCO / ESMO?

Thank you for your comment. As you pointed out, the article should be adjusted based on the established guideline. The guideline for melanoma in Japan is under revision, and the strategy for the administration of BRAF/MEK inhibitors is under discussion. Therefore, we have re-written the sentences you pointed out according to the ASCO/ESMO guideline.

The introduction does not outline a clear aim for the review, and the search strategy (which I found vague [e.g. use of the term ‘mainly’]) appears to reflect a very unstructured nature to process and purpose.

Thank you for your comment. As you and reviewer 1 suggested, we added Table1 to clarify the search strategy.

Abstract

Again, is the abstract in alignment with ASCO / ESMO guidelines for the use of BRAF+MEK inhibitors in melanoma. The aim of the review needs to more clearly stated.

As we described above, we have re-written the sentences that you pointed out according to the ASCO/ESMO guideline.

Pre-clinical studies

There is a lot of important information presented, and the authors should be commended on trying to accumulate this evidence. But is it section truly complete? (e.g. https://www.ncbi.nlm.nih.gov/pubmed/23302800)

Thank you for your comment. We have added sentences to address your comment.

This section requires clarity that the data is pre-clinical – often without independent validation. Which studies have been repeated or the marker investigated in human data? Which have stimulated human clinical trials? Some sentences are written as if the markers are known drivers of resistance within a clinical setting, without validation, I am not sure this is the case.

Thank you for your comments. These pre-clinical studies are just for a future perspective. We have added the sentences in pre-clinical study sesstion.

Phase 1,2 ,3 studies

Again, there is a lot of important information presented. But is it truly complete? https://ascopubs.org/doi/abs/10.1200/JCO.2017.35.15_suppl.TPS9599

A structured search of clinicaltrials.gov may enable clearer and more complete outline of completed clinical trials, as well as trials that are in progress.

Is summarising the Phase 1, 2, 3 studies the major purpose (both completed and underway) of this review?

Thank you for your comment. The purpose of this study is to evaluate the preclinical study as well as phase 1, 2, 3 studies to discuss the future perspective of BRAF inhibitors.

Adverse events

I question if this section belongs in a review centred on outlining clinical studies / trials conducted on BRAF+MEK inhibitors – an entire review could be written on this subject, and there are examples.

Thank you for your comment. We rearranged the section, and adverse events are now included in section 4.4.

Future directions

The summary of human clinical trials that are currently being conducted needs to be expanded – and hopefully the pre-clinical data will provide some interesting context as to why they are being conducted.

Thank you for your comment. We have added further information along the lines suggested.